# Development and Validation of a New Screening Tool with Non-Invasive Indicators for Assessment of Malnutrition Risk in Hospitalised Children

**DOI:** 10.3390/children9050731

**Published:** 2022-05-17

**Authors:** Petra Klanjšek, Majda Pajnkihar, Nataša Marčun Varda, Mirjam Močnik, Sonja Golob Jančič, Petra Povalej Bržan

**Affiliations:** 1Faculty of Health Sciences, University of Maribor, Žitna ulica 15, 2000 Maribor, Slovenia; majda.pajnkihar@um.si; 2Department of Paediatrics, University Medical Centre Maribor, Ljubljanska ulica 5, 2000 Maribor, Slovenia; natasa.marcunvarda@siol.net (N.M.V.); mirjam.mocnik@ukc-mb.si (M.M.); sonja.golobjancic@ukc-mb.si (S.G.J.); 3Faculty of Medicine, University of Maribor, Taborska ulica 8, 2000 Maribor, Slovenia; petra.povalej@um.si; 4Faculty of Electrical Engineering and Computer Science, University of Maribor, Koroška cesta 46, 2000 Maribor, Slovenia

**Keywords:** nutritional risk, pediatrics, undernutrition, validation, computer-based tool

## Abstract

There is no evidence of the most effective nutritional screening tool for hospitalized children. The present study aimed to develop a quick, simple, and valid screening tool for identifying malnutrition risk of hospital admission with non-invasive indicators. A cross-sectional study was conducted. Children`s nutritional baseline using a questionnaire, subjective malnutritional risk, and Subjective Global Nutritional Assessment were assessed on admission. Concurrent validity was assessed using American Society for Parenteral and Enteral Nutrition (ASPEN)and Academy of Nutrition and Dietetics assessment and Subjective Global Nutritional Assessment tool. A new screening tool Simple Pediatric Nutritional risk Screening tool (SPENS) was developed, and sensitivity, specificity and reliability were evaluated. A total of 180 children aged from 1 month to 18 years were included (142 in the development phase and 38 in the validation phase). SPENS consist of four variables and shows almost perfect agreement with subjective malnutritional risk assessment (κ = 0.837) with high sensitivity and specificity (93.3% and 91.3% respectively). Compared with Subjective Global Nutritional Assessment and ASPEN and Academy of Nutrition and Dietetics assessment, SPENS had sensitivity 92.9% and 86.7%, a specificity of 87.5% and 87.0%, and an overall agreement of 0.78 and 0.728, respectively. Due to the fast, simple, easy, and practical to use, screening the SPENS can be performed by nurses, physicians, and dieticians.

## 1. Introduction

Nutritional status affects all aspects of a child’s health, including growth and development, physical activity, and response to serious illness [1]. In hospitalised children with different clinical conditions, adequate nutritional status plays a key role in normal growth processes, responses to concomitant diseases, quality of life, care costs, and survival [2,3]. Malnutrition during the period of illness can interrupt treatment, worsen the child’s existing health condition, and prolong hospitalisation. In addition to failure to thrive [4], malnutrition can have lasting consequences for children, including changes in anthropometric parameters or new emerging diseases [5].

Malnutrition is often unrecognised and untreated in hospitalised children [6]. In previous studies, prevalence rate varies greatly, from 5.1% to 55.6% [7].

The development of malnutrition in hospitalised children is most often influenced by the underlying disease or its worsening [8,9], and further by the presence of chronic diseases [10,11]. The latter can be accompanied by the presence of eating disorders [12] and reduced food intake before or during hospitalisation [13]. The most common gastrointestinal factors of malnutrition include nausea and vomiting, constipation, diarrhoea [8,10] and infections [14]. Surgery, aggressive therapies (e.g., oncology treatment) [15,16] and neuromotor disabilities [17] are also negatively related to nutritional status. The most important clinical indicators of children with disease-associated malnutrition ranked on the basis of the opinions and practices of an international cohort of health professionals are ongoing weight loss, increased losses, increased requirements, low dietary intake and a high-risk condition [18].

Early and timely nutritional screening helps to improve nutritional care, accelerate treatment and recovery, reduce complications, and reduce economic costs in the health system [19] as with other diseases where early diagnosis and timely initiation of treatment are of the utmost importance [20]; therefore, European Society for Paediatric Gastroenterology, Hepatology and Nutrition (ESPHAGAN) and American Society for Parenteral and Enteral Nutrition (AESPEN) are recommending nutritional screening for hospitalised children with paediatric nutritional screening tools (NSTs) which are simple, useful, and cost-effective. Several NSTs for children admitted to the hospital have been developed and validated [7,21]. In the last decade, most of the studies consider Paediatric Yorkhill Malnutrition Score (PYMS), Screening Tool for the Assessment of Malnutrition in Paediatrics (STAMP) and Screening Tool for Risk on Nutritional status and Growth (STRONGkids) [22,23,24]. Due to the individual needs of the different study populations, researchers are continuously developing new NSTs [21]; however, no agreement has been reached on the “gold standard” for the assessment of malnutrition risk [25]. The last point is also one of the reasons that the implementation of routine nutritional risk screening upon hospitalisation has not yet been established in most clinical settings. Among 588 paediatric gastroenterologists and paediatric dietitians from six countries (Australia, Israel, Spain, Turkey, Netherlands, and UK) included in a recent study [18] only 23% reported routine use of NSTs at the hospital (most frequently in Belgium, Netherlands, and UK: 40–50%). The most common approach was assessment of weight changes (85%), followed by the use of growth charts (77–80%).

Low awareness, lack of guidelines or local policy and lack of resources were recognised as the most important barriers to the routine evaluation of disease-associated malnutrition in clinical practice [18].

The development of NSTs modified for each hospital and child diagnosis individually and with excellent reliability, regardless of the person performing the nutrition screening, is suggested [26]. Systematic screening with NSTs is not routinely performed in Slovenian hospitals. A specific NST for identification of the risks of malnutrition in Slovenian children has not yet been developed.

The proposed study focuses on the development and validation of a reliable NST Simple Pediatric Nutritional Risk Screening tool (SPENS) for hospitalised children of all ages (1 month to 18 years), regardless of the child’s disease and the purpose of hospitalisation. The aim of the study is to develop a useful and time-efficient NST that will be used for screening in clinical pediatric practise by nurses, physicians, and dieticians. An implementation of a newly developed SPENS will be the first step towards a systematic routine for nutritional treatment of children in the Slovenian clinical environment; this will enable the identification of hospitalised children with risk of malnutrition, which will further receive appropriate nutritional assessment and timely nutritional interventions.

## 2. Materials and Methods

### 2.1. Design

This study employed a cross-sectional design using a questionnaire, full nutritional assessment, and assessment with Paediatric Subjective Global Nutritional Assessment (SGNA) [27,28,29] tool between 25 May and 16 October 2021. A two-phase exploratory sequential mixed method design was used and made it possible to achieve more comprehensive and richer results than either method independently would have achieved [30]. Qualitative data were the basis for conducting a quantitative data analysis.

### 2.2. Setting and Samples

The setting for this study was one tertiary paediatric clinic in Slovenia. The clinic is the second-largest clinic in Slovenia. As advised [7,31], consecutive sampling was used in the quantitative part of the study. The participants were parents with children from 1 month to 18 years old, who have been admitted to the hospital between May and October 2021 with an expected hospital stay (LOS) of at least 24 hours, regardless of the cause of admission or the child’s medical condition. For the recruitment of children, we followed the eligibility criteria (see Appendix A).

To determine the sample size, we considered the prevalence calculated by the SLOFIT study, where 4.7% to 9.5% of children in the general Slovenian population were malnourished [32]. Based on a 10% prevalence, a sample size of 138 hospitalised children was calculated.

### 2.3. Participants

The sample included 142 parents with children for NST development hospitalised from 25 May 2020 to 21 July 2020 and 38 parents with children hospitalised from 1 September 2020 to 16 October 2020 in the SPENS validation. A total of 180 paediatric and surgery patients aged 1 month to 18 years choose to participate (32.72%).

### 2.4. Measures

The final version of the questionnaire was based on a systematic review [7] and an extensive review of articles related to the causes and consequences of malnutrition in hospitalised children. The questionnaire contained 94 questions from 15 content sets (see Appendix A). Questions used in the form included 277 variables due to several multiple-choice questions. Measurements of anthropometric parameters were recorded numerically.

A post-admission subjective malnutritional risk assessment by a physician included: nutritional history; physical examination (assessment of muscle and subcutaneous fat, detection of swelling and/or ascites); laboratory blood tests; and anthropometric measurements according to ASPEN and Academy of Nutrition and Dietetics recommendations [33]. Techniques of inspection, palpation, percussion and/or auscultation were used [34]. Each individual child was then distributed into one of the five categories according to the risk of malnutrition: high; moderate; low; normal weight; overweight; and obesity.

The SGNA assessment tool was also chosen for identifying malnutrition by a physician. SGNA assessment combines detailed questionnaire about subjective nutrition-focused medical history and complete objective physical examination with an overall ranking which is divided into three categories: well nourished; moderate; and severe malnutrition [29].

### 2.5. Procedure

This study was executed in a two-phase mixed-method design, namely an exploratory sequential design [30]. Research began with the collection and analysis of qualitative data and the production of a questionnaire, followed by a quantitative phase (development and validation of the screening tool) where the initial findings were validated and generalised (Figure 1). The gathering of data and development of a SPENS was conducted from 25 May to 21 July 2020. The SPENS was developed in August 2020. The validation of the tool was performed between the 1 September and 16 October 2020.

#### 2.5.1. Collection and Analysis of Qualitative Data and Development of an Instrument (Questionnaire)

The causes and consequences characteristic of malnutrition in hospitalised children were identified from the literature by an inductive generation of categories [35]. Questions from existing paediatric NSTs were also added [22,24,36,37,38,39,40,41,42,43,44,45,46,47,48,49,50,51,52,53]. The translation of the questions from English to Slovene and vice versa was performed.

The questionnaire was then reviewed by a team of assessors (six nurses, three physicians). Face validity and the appropriate use of standardised professional language [54] was assessed. For questions described as partially understandable by the assessors, a better formulation of the question was suggested.

#### 2.5.2. Gathering Data for SPENS Development

Parents of every child admitted to the clinic between 25 May 2021 and 21 July 2021 were invited to allow their child to participate in the development study.

Each day of the study period, a list of admissions for the current day was reviewed. After exclusion criteria were applied, a list of children for malnutrition risk assessment review was prepared and distributed to one research nurse who served as an interviewer. A post-admission subjective malnutritional risk assessment and SGNA assessment were evaluated in each of the first 80 admitted children by two of three physicians and in the next 62 admitted children by one physician. Consecutively inter-rater reliability was calculated. The Kappa value varied between 0.73 (0.39, 1) and 1 (1, 1) which indicated substantial to almost perfect agreement.

Patient recruitment acted as a secondary examination after admittance. No specialised interventions were performed, only descriptive data were gathered. A post-admission interview using a questionnaire and detailed anthropometric measures was carried out by a research nurse. Parents and/or children answered questions about the factors and consequences of malnutrition. Children under 4 years of age were expected to answer only basic questions, all other information was provided by their parents. The results of the interview were not known to the physicians and the results of nutrition risk and SGNA assessment were hidden from the research nurse.

Detailed anthropometric measures of weight, height and/or length were taken by a research nurse using standardised methods described by the World Health Organisation (WHO) [55]. Body mass index (BMI) by age and sex was calculated using of PediTools [56] and WHO Anthro [57] software. Mid upper arm circumference (MUAC) was measured using a regular flexible plastic tape measure at the mid-point between the acromion and olecranon [41]. Anthropometric data were gathered in Z scores. The WHO Anthro computer program was used to assess the nutritional status of children < 2 years of age [57]. For children ≥ 2 years of age, Centres for Disease Control and Prevention (CDC) 2000 growth curves using a computer program included in the PediTools clinical tools for paediatric practitioners were used [56]. The following indicators of malnutrition were used to determine malnutrition: weight-for-height/length (WFH/L), BMI, height-for-age (HFA), MUAC as recommended by ASPEN and Academy of Nutrition and Dietetics (see Appendix A) [58]. It is advisable to obtain all indicators when assessing malnutrition in children, although only one indicator is needed to diagnose malnutrition [33]. Additionally, for the purpose of calculating nutritional prevalence, values of BMI which define overweight, and obesity as recommended, were also used [59].

Results of individual risk malnutrition score given by physician were given in six risk groups and were consolidated into two risk categories in the development phase of the SPENS: “not at risk” (normal, overweight, and obese risk category) and “at risk” (low, moderate, and severe risk category). SGNA categorization was: well nourished; moderate; and severe malnutrition [29].

#### 2.5.3. Development of SPENS

The first phase of the SPENS development included data cleaning and feature selection. In the second phase, the SPENS was developed using multivariate logistic regression model.

#### 2.5.4. Validation of SPENS

The parents of every child admitted to the clinic between 1 September 2021 and 16 October 2021 were invited to allow their child to participate in the validation study. In the validation phase, the developed SPENS was used for screening in 38 hospitalised children. Screening of children was executed as part of routine examinations during hospitalization by research nurse. A physician also assigned a subjective malnutritional risk assessment to all children who participated in the screening. The same protocol as described in the SPENS development phase was used (Figure 1). The results of the screening were not known to the physician and vice versa. We validated the tool according to recommendations by Klanjsek et al. [7].

For criterion validity each result in children with risk of malnutrition according to SPENS was compared with a reference standard subjective malnutritional risk assessment (n = 38).

For concurrent validity the results obtained by the SPENS were compared with the SGNA assessment tool using chance-corrected agreement (Kappa-statistics) (n = 38). The research nurse performed screening assessment with the developed tool, and the physician performed a nutritional assessment with the SGNA assessment tool and subjective malnutritional risk assessment in the same patients parallel in the same day. Comparison of the two tools (SPENS and SGNA assessment tool) was made with the subjective malnutritional risk assessment.

### 2.6. Data Analysis

Statistical analyses were performed using SPSS software version 28.0 [60] and the R programming language in the RStudio programming environment [61]. Descriptive statistics were used for the presentation of demographic data. Numerical variables were presented with a median (95% CI). Frequencies and percentages were used to describe categorical variables.

The feature selection process in the development phase included two steps. Relationship between risk factors and subjective malnutritional risk assessment was validated by Pearson Chi Square test. Furthermore, the importance of the variables was assessed using the Random Forest (RF) model Importance function, which is based on calculating the Mean Decrease Accuracy of the model [62]. Variables whose Mean Decrease Accuracy metric had a value more than 2 were included in the development of the tool. A manual stepwise multivariate logistic regression (LR) method was used to obtain the model.

The screening tool was validated using the area under the ROC curve (AUC), sensitivity (Se), specificity (Sp), negative (NPV) and positive (PPV) predictive value.

The agreement between the SPENS and subjective malnutritional risk assessment was determined by Kappa (κ) value. κ values were rated with the proposed classification system by Landish and Koch [63]. Se and Sp values were rated as suggested by Bokhorst-de van der Schueren et al. [64] and Klanjsek et al. [7]. The level of statistical significance was set at *p* < 0.05.

### 2.7. Ethical Considerations

Ethical approval was obtained from the Commission of the Republic of Slovenia for Medical Ethics (approval number 0120-329/2016-3 KME 40/07/1). Site-specific approval was obtained for the involved hospital. All participants were recruited after receiving written information and a verbal explanation of the study and obtaining written consent from parents.

## 3. Results

### 3.1. Characteristic of Children in Development and Validation Phase

Between 25 May to 21 July 2021 and 1 September to 16 October 2021, 550 paediatric and surgery children were admitted to the hospital; 180 children (32.7%) were successfully included in the study; 86 (47.8%) were male and 94 (52.2%) females. Median chronological age of our group (n = 180) was 120.62 (108, 142) months, with minimum 1 month and maximum 216 months. The median age of children in SPENS development study was 123 (93, 138) months and in SPENS validation study was 143 (113, 169) months. Children came from a variety of six medical wards. A total of 142 children were included in the development phase and 38 in the validation phase. Sample characterization of the development and validation phase is shown in Appendix A.

### 3.2. Prevalence of Malnutrition of Children in Development and Validation Phase

Prevalence in most of the malnutrition classifications has not varied considerably between the cohorts of children recruited for the development and validation phase (see Appendix A). Prevalence of malnutrition was 40% in the whole sample (40.1% in the development and 39.4% in the validation phase) according to the ASPEN and Academy of Nutrition and Dietetics and 38.4% on the whole sample (38% in development and 39.5% in evaluation phase) according to the subjective malnutritional risk assessment.

### 3.3. Development Phase

Structured questionnaire (n = 277 variables) responses were compared with the classification of nutritional status by a subjective malnutritional risk assessment using chi-squared tests (not shown); this analysis identified 144 significant variables related to malnutritional risk. Additionally, the 144 variables were reduced to 30 most important variables which had Mean Decrease Accuracy > 2. These variables were then used in multivariate logistic regression (LR) analysis. The optimal NST, which includes only four variables is presented in Table 1.

The first two variables include physical examination focused on signs of malnutrition, the third variable includes the child’s rejection of food, and the last one poor weight gain. The first two variables were obtained from the Subjective Global Assessment (SGA) screening tool [42,69]. Variable 3 was identified through an extensive review of the literature and by the inductive generation of categories. Last variable was obtained from Paediatric Nutrition Screening Tool (PNST) [38]. The AUC of SPENS is 0.977 (0.922, 1), Se = 93.3% and Sp = 91.3% with the chosen cut-off value 0.382.

### 3.4. Validation Phase

#### 3.4.1. Criterion Validity

The SPENS was tested on 38 children. It has shown a very good performance (Table 2). Among 15 (39.47%) children who were at risk of malnutrition based on the subjective malnutritional risk assessment, 14 (36.8%) children were classified as at risk of malnutrition with SPENS. Among 23 (60.53%) children who were not at risk of malnutrition based on the subjective malnutritional risk assessment, 21 (55.26%) children were classified as not at risk of malnutrition with SPENS. The AUC on the evaluation set was 0.977 (0.922, 1), with a sensitivity of 93.3% and a specificity of 91.3%. The developed tool also boasts high positive and negative predictive value (PPV = 87.5%, NPV = 95.5%); furthermore, high agreement between the SPENS and predictions with subjective malnutritional risk assessment (κ = 0.837 (0.659, 1.014)) indicates good reliability.

#### 3.4.2. Concurrent Validity

When comparing the SPENS with SGNA assessment as reference method, SPENS had a sensitivity of 92.9% and a specificity of 87.5%. The agreement between these two tools was substantial (κ = 0.78 (0.58, 0.98)). When comparing SPENS with ASPEN as a reference method, SPENS had a sensitivity of 86.7% and a specificity of 87.0%. The agreement of these two tools was substantial (κ = 0.728 (0.474, 0.895)) (Table 3).

## 4. Discussion

The malnutrition prevalence of hospitalised children in this study was 38.4% which is still within the reported range 10.4% to 52.7% of malnutrition in previous studies [6,7]. Child malnutrition is common at hospitalization and may worsen during hospitalization or may be developed a new [6]. Although malnutrition acquired during hospitalization has been shown to be associated with poorer clinical outcomes, longer hospitalizations, and consequently higher treatment costs, it is still underestimated and often unrecognised [70,71].

The SPENS represents the first paediatric NST developed and validated for hospitalised children in Slovenia. The results of this study revealed that the developed tool is reliable for the early detection of malnutrition risk among hospitalised children aged 1 month to 18 years, regardless of the child’s diagnosis and the purpose of hospitalization.

We found that the physical examination of potential visible signs of loss of subcutaneous fat in the face and chest, the child’s refusal/rejection of food, and poor weight gain in the last few months were the most important nutrition risk factor. These four variables included in the SPENS were obtained from initially 277 variables included in the questionnaire through the complex analyzing process in the development phase.

Weight loss in children is shown to be accompanied by a decrease in muscle and fat mass [39,72]. Only four existing screening [23,42,51,69] and one assessment [29] tool includes the assessment of muscle/fat loss in children as an indicator of malnutrition. Physical examination to determine loss of muscle and/or fat mass in children is considered in the literature to be an “overly subjective” indicator [73]; however, physical contact with child’s muscles, bones, and fat provides substantial empirical evidence of malnutrition that is probably more “objective” than asking parents to report exactly what the weight or height of their child is [74]. In practice, both a comprehensive physical examination and an inquiry of food intake are invaluable in diagnosing malnutrition in children [58,75].

Children may have poor eating habits that may continue during formal health education after the age of 18 years [76]. Poor food intake should be associated with fat and/or muscle loss, weight loss, poor body growth and other symptoms [33]. Poor food intake for two to three days can lead to malnutrition in at-risk children, in contrast to well-fed children without disease or other complex medical conditions [73]. We have found that existing NSTs often include nutrition-related issues [22,23,24,36,37,38,40,41,42,43,48,49,52,53,69].

In our developed tool, we ask about poor weight gain over the last few months [38], similar to other NSTs which ask parents for subjective opinion of weight gain such as poor or minimal weight gain in child [23,48]. Other published paediatric NSTs often include anthropometric measurements of body weight [24,37,40,44,45] and height [24,40,44,45] for later use of reference curves or growth tables [24,37,40]. Anthropometric measurements have been found to be routinely poorly performed in hospitals at the time of admission [24]. Existing paediatric NSTs also include ideal body weight [44,45], net weight change [47,50], percentage of weight loss [39,42,49,53,69] and BMI calculation [22,40,47,48,53]. For these measurements and calculations, screening performers need prior education [22], training [24], additional equipment [39], and the final assessment of screening is also influenced by the experience and qualifications of the performer [77]. Due to the above, we have purposely developed a tool that does not include the necessary additional anthropometric measurements and calculations.

In the absence of a gold standard for the assessment of the nutritional status in hospitalised children, subjective malnutritional risk assessment by physician was used as the criterion for developed NST and for the evaluation of its validity [7,40,43]. It has been assumed that a subjective malnutritional risk assessment by a physician is most likely to be accurate, reflecting additional knowledge. The subjective malnutritional risk assessment has also been used as a reference standard in previous studies [24,78]; however, the use of other existing screening or assessment nutritional tools as a reference standard for determining the nutritional status or risk of malnutrition in hospitalised children is not recommended [7].

Measured properties that are important for the usability of a new screening tool include at least the results of sensitivity, specificity, positive and negative predictive values, reliability, and validity [7,54]. For the tool to be truly effective, it should identify those individuals who are really at risk, so the measured values of sensitivity, specificity, and predictive values should be high [21,26,54,79,80,81,82]. Our study demonstrates a strong validity of SPENS, with the ROC analysis indicating the validity of the tool to be excellent when assessed against subjective malnutritional risk assessment. Sensitivity, specificity, positive and negative predictive value of SPENS were 93.3%, 91.3%, 87.5% and 95.5%, respectively. Lu et al. [40] explained that NSTs should have a high sensitivity to minimize the number of false-negative results [82]. Further, sensitivity is more important than specificity, because a false-positive result will only subject the patient to a detailed nutritional assessment, whereas a false negative result can result in an undetected condition [78]. A highly sensitive test is clinically important when identifying a serious but treatable condition like malnutrition, with the main purpose of an NST being to minimize subjects who are at risk of malnutrition being overlooked and not referred for nutritional assessment and intervention [43]; therefore, based on the results, the SPENS is a very reliable NST. As reported in a comprehensive systematic review of 26 validation studies [7], the sensitivity of the tools ranged from 15% to 100% and the specificity ranged between 0% and 96.54%. The use of different reference standards could be the reason for differences among the studies [7,21,26,40,79,81]. Due to the heterogeneity of reference standards, and different limit values, it is almost impossible to compare different NSTs with each other and conclude which is the best [7]; moreover, comparing the relative advantages of different NSTs is misleading, as different tools have been designed for different diagnostic and/or prognostic purposes [83].

Completion of SGNA is lengthy and time-consuming. SGNA is classified as a nutritional assessment aid form and not as an NST, it detects children with already developed malnutrition more than children at risk of malnutrition [81]; nevertheless, many NSTs have been validated with SGNA. In the study of White et al. [38] the sensitivity and specificity for the PNST compared with the paediatric SGNA were fair [7], scoring 77.8% and 82.1%, respectively. The sensitivity and specificity for the SGNA compared with the WHO and CDC 2000 criteria were fair (BMI: Se = 96.5%, Sp = 72.5%; WFA: Se = 85.7%, Sp = 69.7%) or poor (HFA: Se = 46.2%, Sp = 66.5%) [7,38]. SGNA were relatively poor at detecting patients who were stunted or overweight, with a sensitivity and specificity < 67%. Gerasimidis et al. [84] compared STAMP, PYMS, and SGNA to a full dietetic assessment. They found that SGNA poorly identified malnourished children (Se = 15%, Sp = 100%). The agreement between the SGNA and the PYMS were slight (κ = 0.12; 95% CI −0.11, −0.34) and agreement between the SGNA and dietetic assessment were fair (κ = 0.24; 95% CI 0.10, 0.50) [84]. In other studies, SGNA has also been used as a reference standard for the nutritional status of hospitalised children [38,43], which is not recommended [7]. The sensitivity and specificity of the SPENS when compared to SGNA was good (Se = 92.9%, Sp = 87.5%).

Sequential sampling should be used to avoid bias and to make the screening tool suitable for all clinical features and age groups of children [7]. Therefore, we also used sequential sampling in this study. The number of children involved in research for the development and/or validating of NSTs is like other studies [24,41,43,48,49,53].

In some cases, screening performers report that NSTs are time-consuming [40,85], impractical for use in all hospitalised children, and would increase daily workload [86]. More often, it is reported that NSTs are fast [24,37,38,43,50,53] and simple [24,38,43,48,50,52,53]. The results of the screening with developed SPENS are obtained by a quick physical examination of the child and two simple questions that each parent and/or the child knows how to answer; it does not include anthropometric measurements, the use of tables, other necessary calculations, and invasive interventions like the PNST [38] and Nutrition screening tool for childhood cancer (SCAN) [43]. Due to the above, the implementation of our tool should not represent time complications even in the continuous work of health care providers. Compared with Pediatric Nutritional Screening Score (PNSS), STAMP, St Andrews Healthcare Nutrition Screening Instrument (SANSI), Patient-Generated Subjective Global Assessment of nutritional status (PG-SGA), Clinical Assessment of Nutritional Status and the Score (CANSCORE) and SGNA, which are time-consuming [40,81,85], SPENS was efficient, fast (approx. 1–2 min), simple, easy, and practical to use. Screening with PNSS, SANSI and PG-SGA tools took 10 min to complete [40,46,53], while STAMP was completed in 10 to 15 min [79]. PNSS and STAMP require the interpretation of growth charts, SANSI and PG-SGA require prior weight knowledge and BMI and/or weight loss/change calculations and CANSCORE is a scoring system based on nine ‘superficial’ readily detectable signs of malnutrition [51].

An important part of the limitations in the development of tools is the level of reading comprehension, ambiguity, jargon, positive and negative text, and words that each performer can interpret in their own way [54]. In the SPENS, the first two questions contain a concrete descriptive explanation of what child’s body part should look like for the screening performer to confirm the answer as positive.

Similarly to the Pediatric Digital Scaled MAlnutrition Risk screening Tool (PeDiSMART) [37], SPENS can be integrated into an existing computer program at the clinic, which employees already used to manage the treatment and care interventions of hospitalised children; this would allow for a quick calculation, ease of use and time savings for the final screening result. 

Due to the simplicity of the SPENS, we believe that the screening performer does not need prior education or training like in the case of the NST PYMS [22,84], STAMP [24], PeDiSMART [37] or SANSI [53]. To overcome the usual barrier to performing screening in a clinical setting, it is important that the introduction of NSTs into the clinical setting does not require special training on its use and interpretation, and that its completion does not take much time [43].

According to ASPEN’s four principles, NSTs should include at least the first three [87]. The SPENS does include the first three principles, but not the fourth principle related to “Disease severity”. The latter is also not included in CANSCORE [51], SANSI [53], PNST [38], Paediatric Nutrition Rescreening Tool (PNRT) [41], Infant Early Nutrition Warning Score (iNEWS) [52] and two NSTs designed for children with cystic fibrosis [47,48].

We suggest that the nutritional screening with the SPENS is performed directly at admission or in the first 24 hours after the child is admitted and is repeated weekly during the child’s hospitalization. Other authors of NSTs similarly define the time of screening [22,23,37,38,40,43,49,50,53]. Continuous nutritional screening of the child during hospitalization helps to identify those whose nutritional status is deteriorating [33].

First, the main limitation of this exploratory study is the relatively small sample. During the SARS-CoV-2 pandemic, the number of hospitalizations was lower, researchers’ access to the clinic was reduced due to measures, and some parents refused to participate due to fear of COVID-19 infection.

Second, the inter-rater reliability of the tool in yielding the same patients by different assessors was not assessed. And the intra-rater reliability of the tool in the same patients by the same assessor on two occasions (within 24 h period) also was not assessed. Due to SARS-CoV-2 pandemic, restricted measures at the clinic and staffs work overload also produced some limitations. Further studies will focus on the inter and intra-rater reliability, validity, and effectiveness of SPENS in larger number of hospitalised children.

Last, our study was a single-centre case study. A multicentre prospective cohort study would allow the cross-validation of the developed tool in a more diverse demographic.

According to the recommendations [7], the published NSTs are not completely valid, reliable, useful, and acceptable for patients and screening providers. Further research is needed to confirm the applicability of each existing screening tool in the clinical setting [7,81] while further research, refinement and development of the tools are needed.

## 5. Conclusions

The results of our study justify the introduction of screening to determine the risk of malnutrition in hospitalised children in regular clinical practice. The SPENSs validation results are very high, which means that only a few more steps of modification (integration into the clinic’s computer program) would be needed to get the tool ready for routine use in the clinical setting. SPENS is simple, fast, easy, and practical to use; it can be performed by nurses, physicians, and dieticians without special training, and does not require any anthropometric measurements and is not specific for any disease and age of a child. SPENS also includes the first three ASPEN’s principles.

## Figures and Tables

**Figure 1 children-09-00731-f001:**
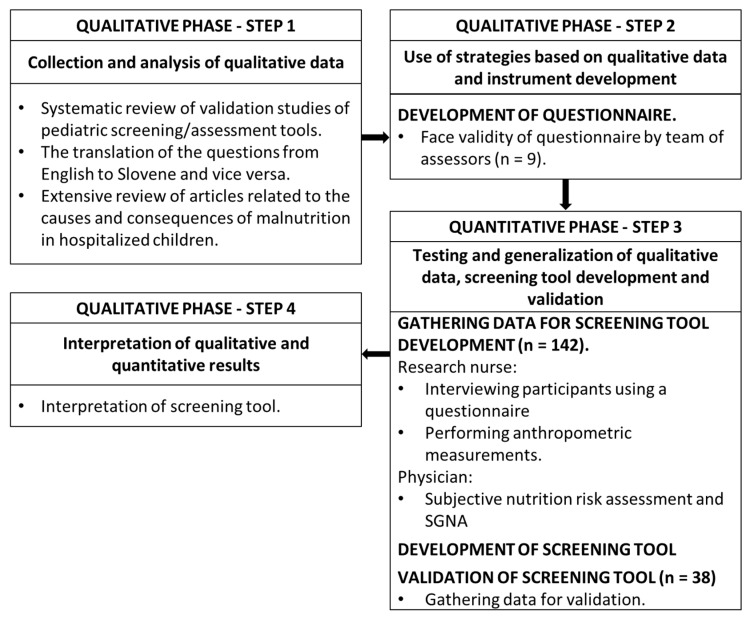
Visual diagram of an exploratory sequential design with the course of the study.

**Table 1 children-09-00731-t001:** Variables included in the SPENS.

.	Variables	Coefficient	Standard Error	Z Score	Odds Ratio (95% CI)	*p*
	(Intercept)	−3.6347	0.6128	5.931		<0.001
1	Loss of subcutaneous fat determined by physical examination under the eyes—(hollowed look, depression and/or dark circles) [42].	2.5491	0.6598	3.864	12.795 (3.695, 50.517)	<0.001 ***
2	Loss of subcutaneous fat determined by physical examination: Ribs, lower back, sides of trunk—Ribs obvious, but indentations are not marked. Iliac Crest is somewhat prominent [42].	2.3982	0.6026	3.980	11.004 (3.522, 38.713)	<0.001 ***
3	Refusal/rejection of food is present [65,66,67,68].	2.4648	0.7031	3.506	11.761 (3.193, 52.357)	<0.001 ***
4	Has a child had poor weight gain over the last few months [38].	1.2805	0.5895	2.172	3.598 (1.135, 11.752)	0.030 *

*p*: statistical significance; * *p* < 0.05; *** *p* < 0.001.

**Table 2 children-09-00731-t002:** Validation of SPENS with the subjective malnutritional risk assessment.

Subjective Malnutritional Risk Assessment	SPENS		
	Not At-Risk	At Risk	Total (n)
Not at-risk	21	2	23
At risk	1	14	15
Total (n)	22	16	38
κ value (95% CI)	0.837 (0.659, 1.014)		
AUC (95% CI)	0.977 (0.922, 1)		
Sensitivity (%)	93.3		
Specificity (%)	91.3		
PPV (%)	87.5		
NPV (%)	95.5		

κ: Kappa value, AUC: Area under the ROC curve, NPV: Negative predictive value, PPV: Positive predictive value, SPENS: Simple PEdiatric Nutritional risk Screening tool, n: number, CI: Confident interval, %: percent.

**Table 3 children-09-00731-t003:** Comparison of the developed screening tool with the other published criteria.

Developed Screening Tool	SGNA Assessment			ASPEN and Academy of Nutrition and Dietetics Assessment		
	Not At-Risk	At Risk	Total (n)	Not At-Risk	At Risk	Total (n)
Not at-risk	21	1	22	20	02	22
At risk	3	13	16	3	13	16
Total (n)	24	14	38	23	15	38
κ value (95% CI)	0.78 (0.58, 0.98)			0.728 (0.474, 0.895)		
AUC (95% CI)	0.912 (0.799, 1)			0.868 (0.739, 0.997)		
Sensitivity (%)	92.9			86.7		
Specificity (%)	87.5			87.0		
PPV (%)	81.3			81.3		
NPV (%)	95.5			90.9		

κ: Kappa value, AUC: Area under the ROC curve, NPV: Negative predictive value, PPV: Positive predictive value, ASPEN: American Society for Parenteral and Enteral Nutrition, n: number, CI: Confident interval, %: percent.

## Data Availability

The data supporting this study’s findings are available from the corresponding author upon reasonable request.

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
