# Peer review of "Development and Validation of a New Screening Tool with Non-Invasive Indicators for Assessment of Malnutrition Risk in Hospitalised Children"

_children, 2022, doi:10.3390/children9050731_

Round 1

Reviewer 1 Report

I do appreciate the efforts of the authors to provide such a described report on the validation of a new screening tool of malnutrition in children hospitalized. I have only a few minor comments to be addressed with the final aim of improving the current version of the paper:

1. The text is detailed; however, it is not easily readable in certain sections (especially, in the methods). I would recomment to shift as much information as possible to the supplemental material, just to let the reader to focus on the clinical significance of the manuscript. 

2. I would spend some additional effort in explaining, even with a few sentence, how this screening tool differ from the already existing ones (i.e. the STRONGkids) and why a clinician should prefer it.

3. English language editing is required throughout the paper.

4. Please update references. More than 80% of references are date back more than 8-10 years. Hereby some appropriate studies that I would add to the references: Nutrients 2020;  doi: 10.3390/nu12082413; Health Serv Res Manag Epidemiol 2021;. doi: 10.1177/23333928211064089

Reviewer 2 Report

The manuscript entitled "Development and validation of a new screening tool with noninvasive indicators for assessment of malnutrition risk in hospitalized children" is an interesting study that could have an important role in the assessment of malnutrition risk. The study is well structured, but the introduction part needs to revise.

My few comments on the manuscript:

  • The introduction is short and general. I am recommending adding more information. For example, it would be interesting to see about other countries' procedures. Furthermore, if there are other existing validated measures worldwide, please introduce those.
  • Another important piece of information would be to explain more aspects of nutrition. On line 96 the Author claims that the "final version of the questionnaire was based on a systematic review" Please add some information on these in the introduction.
  • Subchapter " Development of screening tool" seems to belong to the "data analysis
